# Smoking and COVID-19: Adding Fuel to the Flame

**DOI:** 10.3390/ijms21186581

**Published:** 2020-09-09

**Authors:** Vivek K. Kashyap, Anupam Dhasmana, Andrew Massey, Sudhir Kotnala, Nadeem Zafar, Meena Jaggi, Murali M. Yallapu, Subhash C. Chauhan

**Affiliations:** 1Department of Immunology and Microbiology, School of Medicine, The University of Texas Rio Grande Valley, McAllen, TX 78504, USA; vivek.kashyap@utrgv.edu (V.K.K.); anupam.dhasmana@utrgv.edu (A.D.); andrew.massey@utrgv.edu (A.M.); sudhir.kotnala@utrgv.edu (S.K.); meena.jaggi@utrgv.edu (M.J.); murali.yallapu@utrgv.edu (M.M.Y.); 2South Texas Center of Excellence in Cancer Research, School of Medicine, University of Texas Rio Grande Valley, McAllen, TX 78504, USA; 3Department of Pathology, University of Washington, Seattle, DC 98195, USA; nzafar@uw.edu

**Keywords:** COVID-19, ACE-2, SARS-CoV-2, α7-nAChR, smoking, vaping, e-cigarettes, hooka and cytokine storm

## Abstract

The coronavirus disease 2019 (COVID-19) pandemic, an infection caused by the severe acute respiratory syndrome coronavirus (SARS-CoV-2), has led to more than 771,000 deaths worldwide. Tobacco smoking is a major known risk factor for severe illness and even death from many respiratory infections. The effects of smoking on COVID-19 are currently controversial. Here, we provide an overview of the current knowledge on the effects of smoking on the clinical manifestations, disease progression, inflammatory responses, immunopathogenesis, racial ethnic disparities, and incidence of COVID-19. This review also documents future directions of smoking related research in COVID-19. The current epidemiological finding suggests that active smoking is associated with an increased severity of disease and death in hospitalized COVID-19 patients. Smoking can upregulate the angiotensin-converting enzyme-2 (ACE-2) receptor utilized by SARS-CoV-2 to enter the host cell and activate a ‘cytokine storm’ which can lead to worsen outcomes in COVID-19 patients. This receptor can also act as a potential therapeutic target for COVID-19 and other infectious diseases. The COVID-19 pandemic sheds light on a legacy of inequalities regarding gender, racial, and ethnic health disparities associated with active smoking, thus, smoking cessation may help in improving outcomes. In addition, to flatten the COVID-19 curve, staying indoors, avoiding unnecessary social contact, and bolstering the immune defense system by maintaining a healthy diet/living are highly desirable.

## 1. Introduction

Recently, a severe acute respiratory syndrome coronavirus 2 (SARS-CoV-2) outbreak spread is noticed across the world that is assumed to be originated from Wuhan, China. Figure 1 depicts a graphical representation of top ten infectivity countries from 1st February to 1st August 2020 (data derived from Worldometer, https://www.worldometers.info/coronavirus/). Coronaviruses are RNA viruses which can be classified into four genera based on their genomic characteristics: α, β, π, and ξ [1]. Based on sequencing and evolutionary data, bats are the currently proposed source reservoir for coronaviruses [2,3]. COVID-19 infections have a high homology (>80%) to severe acute respiratory syndrome coronavirus (SARS-CoV), which was responsible for the acute respiratory distress syndrome (ARDS) outbreak in Guandong Province in Southeast China in 2003. The World Health Organization (WHO) declared that the COVID-19 outbreak is a pandemic on 11 March 2020 [4]. As of 16 August 2020, the disease has spread to over 188 countries and territories around the world and has caused more than 21.4 million cases, resulting in more than 771,000 deaths [5,6]. Furthermore, the Centers for Disease Control and Prevention (CDC) has stated that not all COVID-19 infections are fatal, and an estimated that 40% of patients are asymptomatic; however the possibility of transmission from people with no symptoms is 75% and more than 14 million people have been recovered [7,8]. According to WHO and CDC, the COVID-19 virus typically spreads from person to person by respiratory droplets produced during coughing, sneezing or talking and enters the body via mucosal tissues, including the nasal, oral, upper respiratory tract, and less frequently through the conjunctival mucosa [9]. Furthermore, common symptoms include fever, cough, fatigue, loss of the sense of smell and taste, headache, loss of appetite, muscle and joint pain, sore throat, vomiting, blood coughing, diarrhea, and rash [9].

The latest finding suggests that patients with a greater number of comorbidities experienced a greater disease severity from COVID-19 compared to healthy patients without underlying medical conditions [10]. Similar to avian influenza H7N9 virus and SARS-CoV infections, SARS-CoV-2 is more readily predisposed to cause respiratory failure and death in susceptible patients associated with comorbidity factors, including hypertension, diabetes, cancer, and cardiovascular diseases [10,11,12,13,14]. Every year, more than 8 million deaths are associated to tobacco consumption globally [15]. Smokers are known to experience more respiratory complications than non-smokers, as evident by multiple studies showing that smokers experience increased rates of influenza, bacterial pneumonia, and tuberculosis [16,17,18]. As mentioned in previous studies, patients with chronic obstructive pulmonary disease (COPD) are more susceptible for viral infections and presented worse outcomes [19,20]. Notably, patients with pre-existing COPD and current tobacco use are much more likely to experience poor therapeutic outcome from COVID-19 [21,22]. Early evidence shows that in patients who have a history of smoking, the risk of adverse health outcomes for patients with COVID-19 increases dramatically compared to non-smokers, and is associated with higher rates of admission to intensive care units (ICU), use of ventilators, and leading to death(s) [23,24].

Since smoking is a major risk factor for respiratory infections due to suppressive effect of the immune response, thus a hypothetical link between smoking and worsening COVID-19 can be made [25,26]. Smoking is a well-established risk factor in lung cancer, COPD, and asthma [27], which in turn have been shown to be risk factors for more severe illness in COVID-19 patients [28]. To reduce the incidence and severity of an infectious disease, understanding the impact of host factors, particularly avoidable lifestyle factors such as smoking, may play a vital role. Based on recent literature, the WHO confirmed that smokers may experience severe complications from COVID-19 than non-smokers [15]. Considering the complications resulting from smoking in patients with viral infections, we attempted to highlight research focusing on the association between smoking and COVID-19 outcomes, including disease severity, pathogenesis, potential molecular mechanisms, and possible therapeutic interventions.

## 2. The Epidemiology of Smoking and COVID-19

During the COVID-19 pandemic, there is a quest whether smoking increases the risk of acquiring respiratory infections or the biological influence of nicotine on the SAR-CoV-2 [29,30,31]. No strong evidence exists that support an increased risk of COVID-19 in smokers [32]. However, several indirect studies suggest that this population is at a higher risk for a more severe infection amongst hospitalized patients as evident by increased admission rates into ICU, use of ventilators, and death as compared to non-smokers.

Recent studies demonstrated that 1.4–18.5% of hospitalized COVID-19 adult patients were smokers [32]. Two meta-analyses studies (from China) were released the prevalence of COVID-19 in smoker patients. Emami et al. [33] identified 10 articles which covered 76,993 patients and highlighted an overall smoking history of 7.63% of patients (95% confidence interval (CI) 3.83–12.43%) who were infected with SARS-CoV-2. Another meta-analysis with 5,960 hospitalized COVID-19 patients data suggest that current smoking pooled prevalence of 6.5% (1.4–12.6%) [34].

Several epidemiologic case-control and cohort studies show the strong relationship between smoking and severity on COVID-19 disease and death. Zhao et al. [22] identified seven studies and concluded that the risk of severe COVID-19 (Odds Ratio (OR) = 1.98; 95% CI:1.29–3.05) is nearly doubled in smokers. Zheng et al. [35] analyzed the data from thirteen studies comprising 3027 patients and found that smokers had greater disease progression with COVID-19, (current smoking: OR = 2.51; 95% CI: 1.39–3.32; *p* = 0.0006). Baskaran et al. [36] conducted a meta-analysis in 2019 which included 27 studies and 460,592 participants and found that active smokers (pooled OR = 2.17; 95% CI: 1.70–2.76), and ex-smokers (pooled OR = 1.49; 95% CI: 1.26–1.75) were more prone to develop community-acquired pneumonia compared to patients who had never smoked. A recent meta-analysis published by Patanavanich and Glantz which includes 19 peer-reviewed papers with data on smoking and COVID disease progression (17 from China, 1 from Korea, and 1 from the US) concluded that smoking is associated with more double the odds of disease progression in people with COVID-19 infections (OR = 1.91; 95% CI: 1.42–2.59; *p* = 0.001) [37]. Liu et al. [38] conducted a study in a population of 78 patients with COVID-19 and found a statistically significant association between smoking and COVID-19 severity (OR = 14.28; 95% CI: 1.58–25.00; *p* = 0.018). Yu et al. [39] who reported on a study of 70 patients, a statistically significant (OR = 16.1; 95% CI: 1.3–204.2) in a multivariate analysis examining the association between smoking and the exacerbation of pneumonia after treatment.

In contrast, several other studies could not find statistically significant link between smoking and COVID-19 severity. Chen et al. [40] conducted a single center retrospective observational study in Taizhou, Zhejiang, China which included 145 patients with COVID-19. This study concluded that the more severely ill patients had a lower history of smoking when compared to less severely ill patients (7.0% vs. 11.8%, *p* = 0.57). In a recent study, Dong et al. [41] analyzed data from 11 patients with COVID-19 and concluded that there was a non-significant relationship between smoking and severity of COVID-19. More findings by Kim et al. [42] in a recent Korean cohort study with 28 hospitalized patients with confirmed COVID-19 in which only 5 of the patients out of 27 (18.5%) had any smoking history. Another study by Zheng et al. [43] reported that only 10.9% of all patients in the study were smokers, but 6.7% were smokers in the severe/critically ill group, which is not significantly (χ^2^ = 0.962, *p* > 0.05) lower than of the ordinary group with 14.0%. They further concluded that no evidence exists that smoking protects COVID-19 patients from developing to severe disease. Furthermore, another study comprising 140 patients infected with SARS-CoV-2 in Wuhan, China, showed that only 9 (6.4%) patients had a history of smoking, and 7 of them were past smokers, which suggests that smoking history may not be a risk factor for COVID-19 disease severity [44].

The incomplete patient health histories may impact the significance of hospital-based studies [44]. In an emergency context, smoking history collection is challenging and the severity of the disease is often unclear and inconsistent throughout studies at different institutions and likely to lead to significant sampling bias [44,45]. Characteristics of hospitalized patients can also vary based on the services accessible, admission to clinics, treatment procedures, and likely other considerations not included in the studies. Further, most study analyses are based on unadjusted ORs that were either reported in the studies or calculated to account for age and other confounding factors [24,37]. Some peer-reviewed meta-analyses investigating the association between smoking and COVID-19 were also based on unadjusted ORs, but with few studies included [22,35,46,47]. All these evidences support that smoking leads to greater morbidity and mortality in COVID-19 patients.

## 3. Smoking Modulates ACE-2 in COVID-19

The entry of coronavirus into human cells is a multi-step process involving various distinct domains in spike (S) protein, that mediates viral attachment to the cell surface, receptor engagement, protease processing, and membrane fusion [48]. Previous studies have reported that the S protein binding to human cells was facilitated by ACE-2 receptor in several viruses, including NL63, SARS-CoV, and now SARS-CoV-2 causing COVID-19 [49,50,51,52,53]. Through biophysical assays, Wrapp et al. [54] confirmed that the modified S protein of SARS-CoV-2 has a 10- to 20-fold higher affinity for the ACE-2 receptor than SARS-CoV, which results in a more rapid spreading of SARS-CoV-2. Cigarette smoke has been associated with an increased ACE-2 expression in type-2 pneumocytes and alveolar macrophages, particularly at the apical end of the small airway epithelium as compared to non-smokers including current COVID-19 patients, although the mechanism for this is not clear [55,56,57]. The binding of the virus to ACE-2 cell surface proteins shields it against immune surveillance systems, keeping it attached to the host cell for comparatively longer times, rendering it an effective carrier and prone host for potential infections and further spread. Furthermore, the subsequent engulfment of ACE-2 provides the virus with entry into the host cells network, thereby not only surviving and proliferating but also mutating and altering processes of host evasion. Previous research has indicated that SARS-CoV-2 adhesion on ACE-2 could actually downregulate ACE-2 expression, which leads to an enhanced secretion and activation of enzymes associated with ACE-2 [58]. Moreover, this variability and the dramatic reduction in ACE-2 contribute to severe acute respiratory failure [59]. Several other investigators have reported that smoking is associated with ACE-2 in COVID-19 which induces ACE-2 receptor mediated viral entry [60,61]. Furthermore, in vitro studies indicate that the SARS-CoV can be modulated by a number of proteases such as furin, cathepsin L, and trypsin-like serine proteases including transmembrane serine protease 2 (TMPRSS2), TMPRSS11A, and TMPRSS11D [62,63]. It has been reported that TMPRSS2 and furin play essential roles in proteolytic activation of a broad range of viruses including SARS-CoV, Middle East respiratory syndrome coronavirus (MERS-CoV), and SARS-CoV-2 [60,61,62,64,65]. In addition, it has been shown that S protein priming and cell membrane fusion of SARS-CoV-2 is induced by S protein furin cleavage site putative furin recognition motif (PRRARSV) and TMPRSS2 cleavage site (S1/S2 R682/683/685 and S2 R815) [66,67,68]. In addition, the furin site at the SARS-CoV-2 S1/S2 boundary is unique and essential to highly efficient S protein cleavage and a key determinant of the transmissibility and pathogenicity of the virus [65,66,69]. Interestingly, expression of the TMPRSS2 is upregulated in the small airway epithelium of smokers as compared to non-smoker, but not other proteases TMPRSS11A and TMPRSS11D [70]. Studies have also shown that smoking can lead to increases in androgen hormones, such as testosterone. Moreover, the androgen receptor has been found to increase the expression of TMPRSS2 [71].

Nicotine is known to have an effect on the renin-angiotensin system (RAS) by up-regulating the receptor axis, ACE/angiotensin (ANG)-II/ANG II type 1 and down-regulating the ACE-2/ANG-(1–7)/Mas receptor axis, which may in turn contribute to cardio-pulmonary pathologies [72]. In a recent editorial, it has been proposed that the nicotinic cholinergic system could be implicated in COVID-19 infection because COVID-19 clinical manifestations such as a cytokine storm, may be explained by the cholinergic anti-inflammatory dysfunction [73]. Alpha7 nicotinic acetylcholine receptor (α7-nAChR) can potentially modulate pro-inflammatory cytokine secretion, suppressing a cytokine storm [73,74]. The possible mechanism may be that nicotine exposure upregulated the expression of ACE-2, and induced phospho-S6 ribosomal protein (Ser235/236), Akt1, phospho-Akt (Ser473), phospho-Akt (Thr308), and phospho-p44/42 MAPK (Thr202/Tyr204) in vitro; subsequent gene silencing of α7-nAChR appeared to significantly dampen this response [75] (Figure 2). Furthermore, Leung et al. [76] observed an association between airway epithelial ACE-2, α7-nAChR, and the unique vulnerability of patients with COPD to severe COVID-19. Furthermore, another study observed that the risk of COVID-19 neurological infection can be increased with nicotine exposure by a variety of smoking types based on the known functional interactions between the nicotine receptor and ACE-2. The above evidences suggest that smoking induces the expression of ACE-2 in the respiratory tract and smokers have a higher susceptibility to COVID-19.

## 4. Smoking Modulates Proinflammatory Cytokines in COVID-19

Exposure to tobacco smoke also promotes lung inflammation, which results in increasing mucosal inflammation, inflammatory cytokines, and tumor necrosis factor α (TNF-α) expression, as well as enhanced permeability in epithelial cells, mucus production, and muco-ciliary clearance impairment [77]. When SARS-CoV-2 infects cells with highly expressed surface receptors ACE-2 and TMPRSS2, the active replication and release of the virus causes the host cell to undergo pyroptosis and cytokine IL-1β is released, which was confirmed by an increased serum level in patients with SARS-CoV-2 infection [78,79]. Furthermore, the viruses also activate pro-inflammatory cytokines and chemokines IL-6, IP-10, macrophage inflammatory protein 1α (MIP1α), MIP1β, and monocyte chemoattractant protein-1 (MCP1) in infected patients through the use of various pattern-recognition receptors (PRRs), recognized by alveolar epithelial cells (epithelial, endophilic cells) and alveolar macrophages [79]. These cytokines are markers of T helper 1 (Th1) cell activation which is consistent with the findings in SARS-CoV and MERS-CoV infections [80]. These pro-inflammatory cytokines and chemokines attract monocytes and T lymphocytes from the blood into the infected site [81,82]. In addition, the development of lymphopenia (the pulmonary recruitment of blood cells and the infiltration of lymphocytes into the airways, as well as an upregulated neutrophil/lymphocyte counts) was observed in around 80% of patients with SARS-CoV-2 infection [45,83].

COVID-19 patients may experience an over accumulation of immune cells in the lungs accounting for the excessive production of pro-inflammatory cytokines, known as “cytokine storm”, which eventually damages the lung architecture contributing to further additional severe complications. Smoking increases the severity of COVID-19 associated inflammatory response. In a recent study, COVID-19 ICU patients suffered higher systemic inflammatory response due to increased plasma levels IL2, IL7, IL10, granulocyte colony-stimulating factor (GC-SF), interferon-gamma (IFN-γ)-inducible protein (IP10, CXCL10), MCP1, macrophage inflammatory protein 1 alpha (MIP1A), and TNF-α [79]. In severely affected COVID-19 patients, the proportions of IFN-γ yielding CD8+T and CD4+T cells were enhanced as compared to mild case which lead to ‘cytokine storm’ [44,84,85]. Further, higher risks of in-hospital death from COVID-19 are directly associated with lower counts of T lymphocyte subsets (CD3+, CD4+, and CD8+) and B-cells [86]. Another study showed that elevated inflammatory blood markers, including IL-6, can serve as a predisposing factor for increased mortality from COVID-19, suggesting death from virus-activated ‘cytokine-storm syndrome’ [87]. Moreover, patients who survived with severe COVID-19 infections exhibited CD4+T cells, CD8+T cells, IL-6, and IL-10 within normal values, indicating the an attenuation of the cytokine storm [88].

The WHO states that co-morbidities in smoking patients are related to high levels of COVID-19 deaths and complications [89]. Conversely, a plausible theory may be that the cytokine storm can be more readily activated in a perfectly immunocompetent individual than in smokers with increased development of pro-inflammatory molecules (Figure 3). In these ways, we could conclude that a current smoker’s immune system is more tolerant and less reactive than patients who have never smoked and whose immune system could more readily trigger a cytokine release syndrome which may be related to the high death rate associated with COVID-19. This may lead to a partial understanding of the evidence found in studies conducted so far, identified by the vast majority of COVID-19 hospitalized patients as non-smokers.

## 5. Impact of Active and Passive Smoking on COVID-19

The COVID-19 pandemic continues to evolve and poses a global threat to identify prognostic factors. Among the epidemiological risk factors, the role of concurrent smoking as a risk factor for COVID-19 associated pneumonia is controversial. Leung et al. [60] noted the association of higher ACE-2 expression with COPD and current smokers with putatively important implications for COVID-19 patients. Additionally, current smokers showed a higher ACE-2 gene expression as compared to non-smokers and hypothesized that the elevated ACE-2 expression in smokers might predispose them to an increased risk of SARS-CoV-2 infections [60].

In contrast, more recent studies by Rossato et al. [90] and others revealed that COVID-19 patients had a very low prevalence of smoking, with no major connection between active smoking and fatal disease in COVID-19 patients [33,45,91]. These findings resemble with the China’s COVID-19 clinical reports [10,33,45,47]. However, Leung et al. [60] and others concluded that smoking was a significant risk factor for COVID-19 associated pneumonia, in comparison to consolidated epidemiological data from China [10,45,47]. Interestingly, Guan et al. [45] concluded that no significant association exists between current smoking and severe disease in COVID-19 patients because they excluded a sensitivity analysis (a sensitivity test to see the impact of a single study on the smoking and COVID-19). However, in an updated meta-analysis by Fei Ran Guo [92] it was determined that smoking links to the severity of COVID-19 when this same sensitivity test was applied. Furthermore, Lippi and Henry meta-analysis has demonstrated that there is no link between smoking status and COVID-19 severity [47]. In addition, meta-analysis by Vardavas and Nikitara [23] included five studies and concluded non-significant relationship between smoking and severity of COVID-19. More findings by Hu et al. [93] showed that smoking are associated with unfavorable clinical outcomes of disease (OR = 3.464; 95% CI: 1.18–10.166; *p* = 0.001). Moreover, Ernest Loa and Benoit Lasnier [94] state that the meta-analysis by Lippi and Henry had an erroneously interpreted their results due to the use of null hypothesis significance testing (NHST) approach which is known to have major limitations as discussed in recent high-profile statements [95,96]. Leng et al. [46] also published a report regarding Lippi and Henry’s meta-analysis data, in which he attempts to explain the reason behind their conclusion. The misclassification of smoking due to underreporting in these cohorts may be an explanation. The other possible reason might be that “smokers may be taking medications that may offer some protection against COVID-19 [46]. As noted previously, COVID-19 more strongly affects the older population (>65 years) with co-morbidities, while smoking rates in this age bracket are around 3–5 times lower than in the general population. As a result, in the COVID-19 susceptible subgroups, the baseline smoking rates may also be significantly lower than the general smoking rates of the population [46]. Therefore, we should be especially careful about the messages surrounding smoking and COVID-19, particularly in these fraught times where misinformation is commonly amplified in a vacuum of rigorous evidence [97,98].

## 6. Electronic Cigarette, Vaping, Hooka and COVID-19

The role of smoking in COVID-19 raises a question of whether this is also relates to individuals interested with water pipe smoking and those switching over to electronic cigarettes and “I Quit Original Smoking” (IQOS) devices colloquially referred to as “heat-not-burn” devices in which tobacco products are heated up using a battery-powered heating system, with the supposed advantage of reducing the number of harmful chemicals released [99]. Currently, there is little direct evidence showing the effect of e-cigarette use on COVID-19 outcomes. Certain news outlets have started speculating on links between e-cigarettes and worsening COVID-19, primarily based on evidence suggesting that e-cigarettes use leads to lung damage and also that persons with lung problems have more severe symptoms, then e-cigarettes thought to lead to worse outcomes for COVID-19 patients [100].

A recent study demonstrated that mice exposed to e-cigarette smoke/vaping had more complications from influenza than non-vaping mice [101]. Vaping may have impaired immune responses in mice, but it is not clear whether the study results can be applied to humans and COVID-19 patients [101]. Clinical trials and other studies have demonstrated that these devices are not “safer”, and have caused damage to the airways epithelial and as well as a considerable reductions in transcutaneous oxygen tension in young tobacco smokers, and short-term impaired arterial oxygen tension in heavy smokers [102,103,104,105]. Due to their design, these products are brought to the mouth and face to inhale repeatedly, and many users have an increased urge to cough or expectorate which can increase the transmission of COVID-19. In addition, the aerosols and vapors generated by electronic substance delivery systems could contribute to SARS-CoV-2 transmission [106].

Another way of smoking is a hookah (shisha or waterpipe), a single- or multi-stemmed instrument typically used by multiple people simultaneously. In the United States (US), “hookah bars” have gained popularity in recent years with nearly 2.6 million people smoking hookah products and there also an estimated 100 million hookah users worldwide [107,108,109]. By virtue of their design, hookahs are ideal vectors for viral spreading and may escalate the risk for more severe COVID-19 infections through public use, complex cleaning requirements, and a cold-water reservoir, suitable for SARS-CoV-2 transmission [109]. In addition, hookah smoke contains some harmful chemicals that can damage the respiratory lining and predispose smokers to respiratory infection such as MERS-CoV [110].

Due to the risks of public health posed by transmission of SARS-CoV-2, some countries have already imposed restrictions on hooka use [111]. The usage of e-cigarettes and hookah/waterpipe smoking should be stopped during the SARS-CoV-2 transmission cycle or at least from the start of symptoms. Additional research into these products and their influence on the virulence of COVID-19 is still needed.

## 7. Smoking in Sex Predisposition and Racial Ethnicity in COVID-19

The COVID-19 pandemic has exposed longstanding racial and ethnic health-related disparities, largely driven by socio-economic and environmental factors [112]. COVID-19 epidemiological studies also showed sex-specific differences in incidence and fatality rates with a higher rate in males (2.8%) compared to females (1.7%) also associated with smoking [113,114,115]. In a recent study [115], differences in COVID-19 disease prevalence and severity were associated with sex and smoking. Smoking is associated with ACE-2 which have alleles that offer COVID-19 resistance and clarify a lower rate of mortality for women since they are located on the X chromosome [115].

The sex hormones (testosterone and estrogen) have shown distinct effects on immune function, which can affect immune defense or disease severity. There is also a clear sex-dependent incidence of this disease, with a higher incidence in men than in women, which may be due to the higher rate of smoking among men. Several recent studies indicate that estrogens appear to upregulate ACE-2 in younger children and adult women. This seems to lead to a higher level of ACE-2 than in older adults that could explain the milder COVID-19 in women and children and suggest a protective effect [116,117]. The expression of ACE-2 was more prevalent in Asian men, which could explain the higher prevalence of COVID-19 in this group of patients than in women and patients of other ethnicities [118]. Several studies suggest that there could be a sex predisposition to COVID-19, with men more likely to be affected [44,45,119]. One study found that the distribution was equal between sexes as determined in a cohort of 140 patients in China with COVID-19 [44]. Another study report that more males (67%) were diagnosed as critically ill [119]. In a recent study by Guan et al. [45] which included 552 hospitals across 30 provinces in China with 1099 COVID-19 patients, 58% were men. This sex predisposition can be correlated with the significantly higher incidence of smoking among men than women in China. Cai et al. [55] reported that no significant differences were found in the expression of ACE-2 between Asians and White patients, or male and female or older and younger than 60 years. Interestingly, its expression was considerably higher in current Asian ethnic smokers than in Asian non-smokers, but no disparities between White smokers and non-smokers were recorded. However, existing research does not endorse smoking as a predisposing factor for SARS-CoV-2 infection in men or any subgroup. In a recent study from Zhang et al. [44] only 1.4% of patients were current smokers; similar results were indicated in the study by Guan et al., showing only 12.6% of patients as current smokers [120]. In both studies, comparatively low numbers of active smokers are unlikely to be correlated with the prevalence or severity of COVID-19 as compared to the ratio of male smokers in China (50.5%).

Without stronger evidence of the association between the predominance or severity of COVID-19 in Asian men compared to other subgroups, no firm conclusions can be drawn. ACE-2 expression variability can be further studied in more cases from diverse ethnic and genetic backgrounds globally in order to see whether it leads to altered COVID-19 susceptibility in various subgroups.

There are also variations in lifestyle and diseases, mental and physical, through ethnic groups which may clarify the vulnerability to a severe COVID-19 infection [121]. Several co-morbidity factors are suspected to contribute to the disproportionate impact of COVID-19 among African Americans, including high rates of underlying health conditions like cardio-pulmonary diseases and diabetes, which increases the risk of a more severe illness from COVID-19. As mentioned in one study, smoking is a major cause of these underlying conditions, and African Americans experience higher rates of many chronic conditions linked to COVID-19 and 18.3% of African American adults are current smokers [122]. Smoking among African American men is higher than among African American women (21.8% vs. 15.4%) and this population group is more likely to be exposed to second-hand smoke than any other racial or ethnic group. Moreover, mostly African American dominated US counties have a three-fold higher infection rate and a six-fold higher mortality risk than other White dominated counties [123].

## 8. Does Cessation of Tobacco Consumption Lead to Lower COVID-19 Risk?

Studies show that smokers are more prone to developing COVID-19 and more vulnerable to severe COVID-19 complications. A crucial question both to determine causality and to provide advice and effective interventions to patients is whether the risk of developing a tobacco-associated COVID-19 infection can be reduced after quitting smoking. Unfortunately, there are limited existing data on the impact of smoking cessation during the COVID-19 pandemic. Research data suggest that up to 70% of smokers have an interest in stopping smoking, but only 3–10% can do so on their own [124,125]. Smoking cessation before surgery leads to a rapid reduction in nicotine and carboxyhemoglobin a level in the bloodstream, which also provides smokers an opportunity to engage in long-term smoking cessation [126,127]. Smoking withdrawal for 4 weeks or longer has evidence of a decreased risk for COVID-19 and a lower risk of more severe complications. A recent study by Turan et al. [128] examining 635,265 non-cardiac surgical patients noted that current smokers had a higher likelihood of 30-day mortality (RR = 1.38; 95% CI: 1.11–1.72) and serious post-operative complications such as surgical site infection (OR = 1.30; 95% CI: 1.80–2.43), unplanned intubation (OR = 1.87; 95% CI: 1.58–2.21), pneumonia (OR = 2.09; 95% CI: 1.80–2.43), and septic shock(OR = 1.55; 95% CI: 1.29–1.87). In other meta-analysis by Wong et al. [129] that included 25 studies on short term preoperative smoking cessation and post-operative complications, it was found that a minimum of 4-weeks of smoking cessation before surgery lowered the risk of respiratory complications and that cessation of at least three to four weeks also reduced wound-healing complications when compared to current smokers.

Findings also suggest that the serum half-lives of carbon monoxide and nicotine are approximately 4 h and 1 h, respectively [127]. Another study reported that an average carboxyhemoglobin serum level in smokers is 3.81 ± 2.17 g/dL^−1^ as compared to 2.95 ± 1.33 g/dL^−1^ in non-smokers [130]. It is plausible that an increase in cessation rates could help minimize community transmission of SARS-CoV-2. Evidence shows that various strategies incorporating both pharmacological and behavioral interventions during viral epidemics are better positioned to reduce smoking-related complications by encouraging cessation [131]. Healthcare providers should be involved in offering evidence-based pharmacological and behavioral smoking cessation interventions by remote support if in-person visits are not possible. A recent study shows that varenicline is the most effective smoking cessation pharmacotherapy followed by bupropion and nicotine patches [132]. In a 12-week double-blind, randomized, placebo-controlled clinical trial study of 8144 participants, varenicline improved higher abstinence rates compared with placebo (OR = 3.61; 95% CI: 3.07–4.24), nicotine patch (OR = 1.68; 95% CI: 1.46–1.93), and bupropion (OR = 1.75; 95% CI 1.52–2.01) [132]. Furthermore, bupropion and nicotine patch showed a trend toward higher smoking cessations rates as compared to placebo (OR = 2.07; CI: 1.75–2.45 and OR = 2.15; 95% CI: 1.82–2.54, respectively) [132]. Telemedicine can be employed by doctors to advise smokers about the benefits of smoking cessation. Long-term lockdowns will contribute to and increase the frequency of social distancing, and may exacerbate or even cause mental disorders, which can increase the urge to smoke; within these socially deprived populations, smoking becomes more common, which increases the risk of this population contracting COVID-19. Cessation of smoking is expected to reduce the risk of COVID-19 emerging and severe COVID-19 complications.

## 9. Could Nicotine Be a Therapeutic Option to Lower COVID-19 Risk?

Nicotine is a cholinergic anti-inflammatory agonist that regulates the immune and inflammatory responses of the host [133,134,135,136]. Farsalinos et al. [34] claim that pharmaceutical nicotine should be considered a viable therapeutic option for COVID-19. Changeux et al. [137] have proposed the “nicotinic hypothesis,” where nicotinic acetylcholine receptors may be a potential therapeutic target to reduce SARS-CoV-2 infections and alleviate COVID-19 disease. Previous studies have reported that medicinal nicotine has been used successfully for tobacco addiction and has been tested for, the treatment of neurological disorders in non-smokers [138]. Nicotine also prevented acute lung injury in an animal model of ARDS, inhibited TNF-α expression in airway epithelial cells in vitro and exhibited anti-inflammatory properties in vivo in humans exposed to endotoxins [139,140,141]. Several studies have also reported that nicotine inhibits the production of pro-inflammatory cytokines (TNF-α, IL-1, and IL-6), without inhibiting the production of anti-inflammatory cytokines such as IL-10 [133,142,143]. These effects have been shown to protect against cytokine-mediated diseases such as sepsis and endotoxemia, which can lead to organ damage or even death as from a ‘cytokine storm’, the main culprit in COVID-19 [81,144,145,146]. Since expression of IL-6 has been considered a predictor of mortality, it also act as a pharmacological target, tocilizumab (TCZ) has already been studied in clinical trials in an effort to treat patients with severe COVID-19 by neutralizing these key inflammatory mediators [87,144,146,147,148,149].

Furthermore, smoking and nicotine-induced downregulation of ACE-2 was identified prior to the COVID-19 pandemic [72,150]. However, more recent findings indicate that ACE-2 is upregulated in this disease [57,60,120]. This recently identified upregulation of ACE-2 is presumably a protective mechanism against tissue damage caused by SARS-CoV-2. ACE-2 has been previously identified to protect mice from developing ARDS [59,151,152]. Based on previous SARS research evidence, SARS-CoV-2 capable to induce an abrupt reduction in ACE-2, which is thought to be related to increased morbidity and tissue damage [153]. Therefore, ACE-2 upregulation, while apparently paradoxical, will potentially defend patients from severe disease and lung injury [154].

However, these results do not definitively answer the questions surrounding the effects of nicotine or smoking on ACE-2. They can also show the complex equilibrium between angiotensin-converting-enzyme (ACE) and ACE-2, which can shift constantly based on stressors and triggers. Recent study report that local or systemic infection may lead acute lung injury and vasoconstriction via involvement of ACE, angiotensin I and II system in various respiratory diseases including COVID-19. In the same process, ACE-2 has relation with group of angiotensin as a regulatory enzyme which convert the angiotensin II to angiotensin-(1–7) and have role in modification of angiotensin II into angiotensin-(1–9). This regulation will be attenuating the effects on vasoconstriction, sodium retention, and fibrosis induced by angiotensin [153,155]. There is still some uncertainty about how nicotine affects the progression COVID-19 via the renin-angiotensin-aldosterone axis and how ACE-2 and α7-nAChR receptors interact. More concrete evidence is required before medicinal nicotine can be recommended for COVID-19. It is critical to distinguish between cigarette smoking and medicinal nicotine for the prevention and treatment of COVID-19, because smoking cannot play a therapeutic role given the weight of evidence showing its harm, both in general and potentially in COVID-19.

## 10. Bolstering Defenses against COVID-19

At present, considering the terrible COVID-19 pandemic, there is an urgent requirement to explore the effective treatment approaches against COVID-19. Using various computational tools, genetic analyses, and protein modeling, several therapeutic strategies have been proposed using pre-existing drugs repurposed for SARS-CoV-2 therapy in an attempt to bypass the rigor of clinical trials required for novel therapeutic agents [156,157,158,159,160]. As a result, various potential targets, including the viral S protein, ACE-2, TMPRSS2, 3C-like protease (3CLpro), RNA-dependent RNA polymerase (RdRp), and Papain-like protease (PLpro) have been identified for repurposing pre-existing antiviral drugs and new small molecules that are under development against SARS and other coronavirus infections. Here, we will focus on the research progress of chloroquine, hydroxychloroquine, remdesivir, and lopinavir/ritonavir (Figure 4). These drugs can be used alone or in combination to combat against the virus.

The protease inhibitors, lopinavir and ritonavir (anti-retroviral drugs) have shown effectiveness against human immunodeficiency virus 1 (HIV-1) infection, SARS-CoV, MERS-CoV, and SARS-CoV-2 viruses in in vitro susceptibility models [49,161,162]. Protease inhibitors effectively inhibit the 3CLpro enzyme, thus posing a potentially potent therapeutic agent for controlling SARS-CoV-2 infection. To evaluate the efficacy of lopinavir/ritonavir for SARS-CoV-2 infection, a controlled clinical trial was conducted on 134 confirmed patients with novel coronavirus pneumonia. As indicated in a recent study, lopinavir and ritonavir neither improve symptoms nor affected the conversion time in respiratory tract samples [163]. Cao et al. [164] also reported that there was no observed benefit with lopinavir–ritonavir treatment compared to standard care in patients with severe COVID-19. A recent study show that lopinavir–ritonavir co-therapy along with another anti-influenza drug, oseltamivir, was reported to result in complete recovery after showing signs of COVID-19 related pneumonia [165,166]. SARS-CoV-2 and SARS-CoV RdRp share 96% sequence identity; this implies that inhibitors effective against SARS-CoV could have similar inhibitory effects against SARS-CoV-2 [167]. In addition, nucleoside analogs of adenine or guanine derivatives can be used against RdRp to impede viral RNA synthesis. Remdesivir (GS-5734), is a phosphoramidate prodrug of an adenine derivative, and which was used in treatments against Ebola, SARS-CoV, and MERS-CoV, as it has the potential to outcompete the proofreading ability of coronavirus exonuclease, and carries a high genetic resistance barrier [36,43,45,101]. Recent in vitro studies have shown that remdesivir effectively inhibits SARS-CoV-2 (EC50 in Vero E6 cells = 0.77 μM [11,156]. Earlier studies implied that remdesivir exhibited broad-spectrum activity against SARS-CoV-2 and similar viruses (including SARS-CoV and MERS-CoV) [156,168].

Favipiravir (T-705), is a nucleoside analogue that inhibits RdRp of RNA viruses, such as influenza virus, Ebola virus, flavivirus, chikungunya virus, norovirus, and enterovirus [169]. Another recent study has shown that Favipiravir (EC50 in Vero E6 cells = 61.88 μM) against SARS-CoV-2 and blocked viral infection [156]. A clinical study of favipiravir has shown a promising clinical efficacy in treating SARS-CoV-2 and its possibility to be safely included in future treatment plans, as it has shown both efficacy and bio-availability of the drug [170].

In addition, ribavirin and azvudine have shown some promising in vitro results for their efficacy against SARS-CoV-2, and it’s in vivo efficacy is currently being tested in three ongoing clinical trials [171,172].

Chloroquine and hydroxychloroquine are currently licensed for malaria, and autoimmune diseases such as lupus, blood disorder porphyria cutanea, rheumatoid arthritis and many countries have extensively used for COVID-19 treatment [173,174]. Researchers have found that Chloroquine and hydroxychloroquine have in vitro inhibitory effect against several viruses in including SARS-CoV and SARS-CoV-2 [173,175]. Based on these results, chloroquine and hydroxychloroquine are currently promoted for treatment of hospitalized COVID-19 patients in several countries [176]. A chloroquine-based drug was documented to inhibit SARS coronavirus fusion with the cells, by acidifying lysosomes and thereby inhibiting pH-low cathersin to optimally cleavage the S protein SARS-CoV-2 [175]. In addition, chloroquine has been shown to disrupt terminal ACE-2 glycosylation in the Golgi system, thus inhibiting viral penetration into host cells [177]. However, neither chloroquine nor hydroxychloroquine have been used in clinical setting to prevent COVID-19, because their administration is associated with significant adverse effects, such as loss of vision, nausea, stomach problems, and potential cardiovascular issues. Chloroquine and hydroxychloroquine act by accumulating in lymphocytes and macrophages and by reducing the secretion of proinflammatory cytokines, and/or by activating anti-SARS-CoV-2 CD8+T-cells [177]. These antimalarial agents have shown in vitro efficacy, either alone or in combination with azithromycin, against SARS-CoV-2 according to the results of several recent studies [178,179]. However, these results have been challenged by new trials with no substantial advantages from hydroxychloroquine administration even though antimalaria drugs are being tested in more than 30 randomized controlled trials [172].

Virus-induced immune responses leading to cytokine storm syndrome and hyperinflammation are associated with ARDS which can result in multiple organ dysfunction or death [149] Immunosuppressants along with antivirals play a crucial role in counteracting severe SARS-CoV-2 infection [149]. Several other immunomodulators and anti-inflammatory drugs are also being applied in clinical trials [172].

## 11. Public Health Announcement

Policies to reduce excessive tobacco consumption, including tobacco-free public places and the protection of people from second-hand smoke as per Article 8 of the WHO Framework Convention on Tobacco Control, and its guidelines, could reduce the risk of more advanced cases of COVID-19 as caused by tobacco exposure. It is important for smokers to recognize that excessive tobacco consumption use can delay or negatively impact COVID-19 treatment and that reducing high-risk tobacco consumption could be a method of COVID-19 prevention.

In an effort to flatten the curve and minimize the spread of this disease, it is recommended to follow social distancing strategies, practice thorough hand washing for at least 20 s, regularly clean highly-touched surfaces with disinfectants, wear face masks in public areas, engage in proper respiratory hygiene (e.g., covering coughs and sneezes), and to avoid touching the eyes, nose, or mouth with unwashed hands [180]. The following strategies for public health could help reduce high risk smoking during COVID-19. (1) The clinical strategies of smoking screening and counseling, encouraged to quit, should be provided with advice, support, and pharmacotherapy; (2) Regulate tobacco/hookah bar outlets; (3) Increase tobacco taxes and prices; (4) Enhance enforcement of laws prohibiting sales to minors, advertising, promotion and sponsorship of tobacco/tobacco products; (5) Stop targeting vulnerable communities by tobacco industry; and (6) Offering and encouraging for quitting tobacco or tobacco related product consumption.

## 12. Research Needs

Although smoking is a well-established risk factor for the development of various respiratory infections ranging from the common cold to more severe diseases (e.g., MERS and, SARS), there is a little information regarding how smoking use affects COVID-19 disease progression, severity, and treatment delivery [181]. Therefore, the most compelling and urgent research need for the nicotine and tobacco research community should be to investigate the role of tobacco in the current COVID-19 pandemic through the use of well-designed population-based studies that control the age and relevant risk factors of COVID-19. International collaboration between different health organizations is necessary to ensure that more accurate data can be identified, and analyses should focus on the role of this association in viral contamination, illness severity, recovery rates, impact of comorbidities, and so on. More research is needed to examine the interactions between tobacco smoke, SARS-CoV-2, ACE-2, TMPRSS2, and their genes, including the validation of in vitro computational data and in vivo evidence. We need data from symptomatic COVID-19 smokers to determine the immediate and the short-term benefit of smoking cessation. Unfortunately, there are still considerable racial/ethnic disparities in smoking susceptibility over time, and their role in COVID-19 should be explored extensively and interventions are needed to combat this issue. Anecdotal stories from news providers about the effect of nicotine on COVID-19 treatment are intriguing, but rigorous scientific investigations are required to measure the effect accurately. Data on alternate delivery mechanisms for nicotine and them risk/benefit ratio for COVID-19 are also needed. There is no direct link between identified so far between the e-cigarettes/vaping use and risk of contracting COVID-19.

Therefore, we urgently need further research on these products and its impact on virulence of COVID-19.

## 13. Conclusions

The COVID-19 pandemic has spurred a global public health crisis. The studies reviewed here clearly indicate the complexity and multi-factorial etiology of smoking in relation to COVID-19. Epidemiological meta-analyses findings suggest that active smoking is significantly linked with the risk of more severity of COVID-19. Mechanistically, exposure of smoking elevated ACE-2 and activated ‘cytokine storm’ and their associated gene; this may also a possible potential therapeutic target for SARS-CoV-2. Cessation of smoking is likely to decrease the risk of COVID-19 as well as the likelihood of developing more severe complications. Unfortunately, health disparities still exist, as African American and Latino/Hispanic patients have been shown to contracting SARS-CoV-2 at higher rates and are more likely to die.

## Figures and Tables

**Figure 1 ijms-21-06581-f001:**
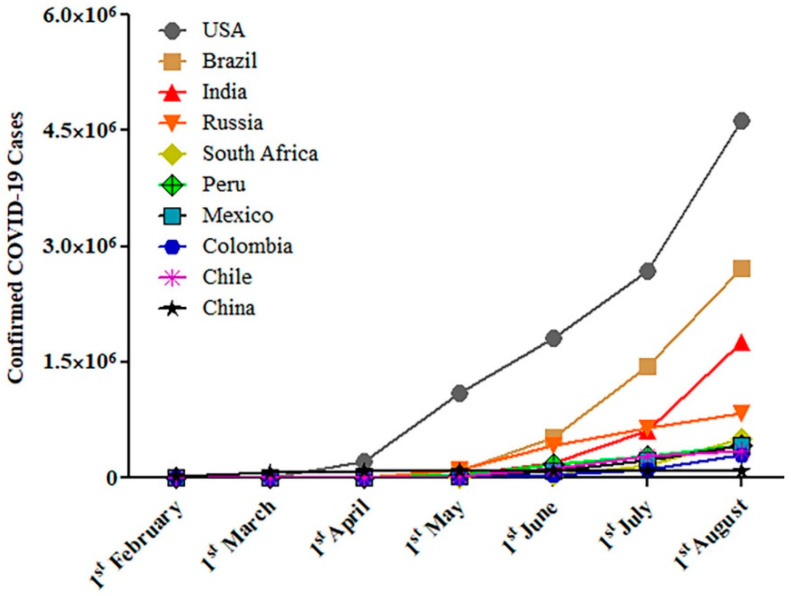
Reported confirmed COVID-19 cases over time in top ten highly infected countries. Data accessed on 15 August 2020, using Worldometer, https://www.worldometers.info/coronavirus/.

**Figure 2 ijms-21-06581-f002:**
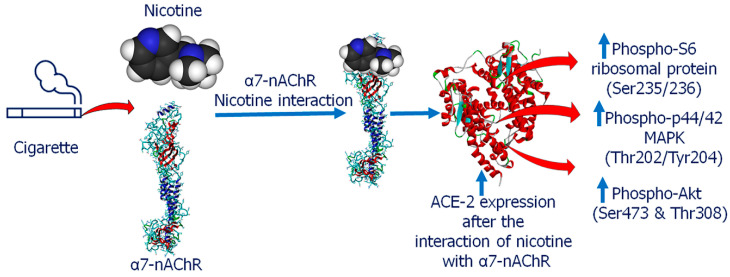
Schematic representation of SARS-CoV-2-driven signaling pathways induced by smoking. Smoking component has tendency to interact with α7-nAChR, this interaction directly stimulates the ACE-2 expression level, which upregulated phospho-S6 ribosomal protein (Ser235/236), phospho-p44/42 MAPK (Thr202/Tyr204) and phospho-Akt (Ser473 and Thr308).

**Figure 3 ijms-21-06581-f003:**
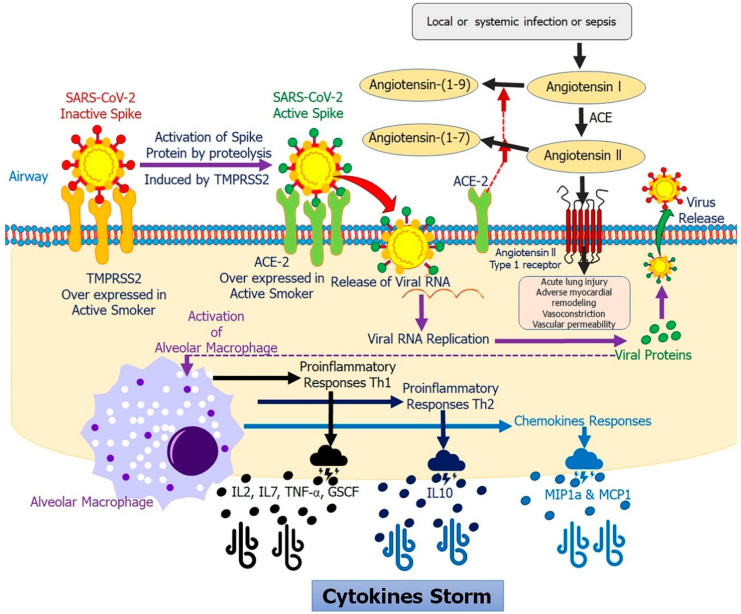
Chronology of events of SARS-CoV-2 infection induced by smoking: When SARS-CoV-2 interacts with TMPRSS2, it activates the S protein by proteolysis. This activated S protein is intended to be attached to and infect cells expressing the surface receptor ACE-2. This activation triggers viral replication and leads to virus release from infected cells. The release virus and their viral proteins also induces molecular events. Once being detected by nearby alveolar macrophages, a cascade is activated leading to the excessive production of pro-inflammatory cytokines and chemokines including IL-2,7, 10, TNF-α, GS-CF, macrophage inflammatory protein 1α (MIP1α), MIP1β and MCP1. These proteins attract other immune cells and promoting further inflammation which establishes a cytokines storm.

**Figure 4 ijms-21-06581-f004:**
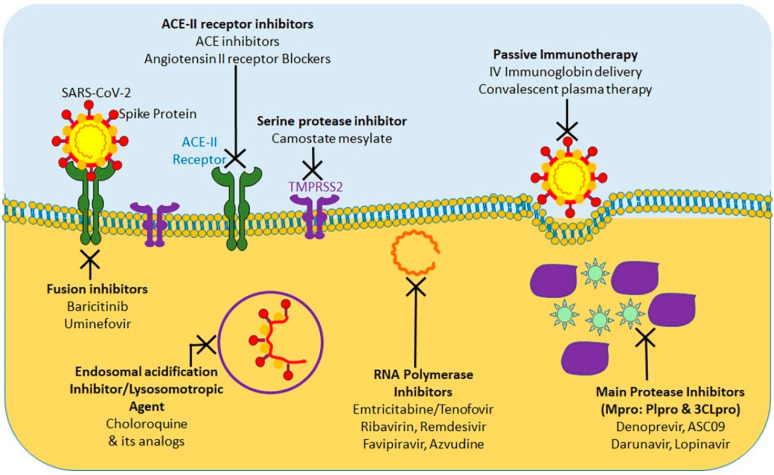
Summary of currently tested potential therapeutic agents targeting different steps of SARS-CoV-2 life cycle.

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
