# Peer review of "Smoking and COVID-19: Adding Fuel to the Flame"

_ijms, 2020, doi:10.3390/ijms21186581_

Round 1
Reviewer 1 Report
Timely and interesting topic. The structure and writing are well done. The figures/schematics look great. The findings described are detailed and interesting. The only question I have that authors should touch more on the contradictory data suggesting that there is an inverse relationship between smoking and COVID severity. May be worth having a section specific to discussing this observation and citing work related to this result.
Author Response
Reviewer #1:
Comment: Timely and interesting topic. The structure and writing are well done. The figures/schematics look great. The findings described are detailed and interesting. The only question I have that authors should touch more on the contradictory data suggesting that there is an inverse relationship between smoking and COVID severity. May be worth having a section specific to discussing this observation and citing work related to this result.
Response: Thank you very much for your suggestion. We have also focused as much possible on contradictory data suggesting that there is an inverse relationship between smoking and COVID severity in different sections.
Reviewer 2 Report
Kashyap and colleagues summarize current literature related to COVID-19 and the impact of smoking on disease severity. The authors highlight studies that associate smoking with changes in ACE-2 expression and pro-inflammatory cytokine responses. The authors further outline a case against smoking, as a countermeasure to the pandemic. This is an important topic relevant to the current SARS-CoV-2 pandemic. Understanding behavioral influences in the risk of COVID-19 will be important for combating disease.
- Section 1. Introduction
- Lines 35-36. Include specifics on the date or time-frame and countries that saw a dramatic rise in cases.
- Line 41. Include the specific province (Guandong Province in Southeast China) in place of “Asia.”
- Line 44. Remove the phrase “and more than 5.23 million people have recovered.” Include this phrase in a new sentence that could state that not all COVID-19 cases are fatal and include the percentage of cases that are asymptomatic.
- Line 47. Replace “nose, mouth” with nasal and oral. End the sentence with less frequently conjunctival mucosa and begin a new sentence starting with “Common symptoms include…” The authors could also include additional symptoms specific to SARS-CoV that includes loss of taste and smell
- Line 50., revise “compared to previously health patients” to read as “compared to healthy patients without underlying medical conditions.”
- Line 51-52. Revise influenza, H7N9 to “avian influenza H7N9 virus”
- Line 57-58. Revise sentence to read as, “Patients with COPD are more susceptible to viral infections, exacerbated inflammation, and loss of lung function compared to individuals with…” What is meant by “those with existing exacertabing disease conditions”?
- Section 2. The epidemiology of smoking and COVID-19:
- This paragraph lists case reports for different studies without synthesizing the findings for the reader. The data reported on lines 79-98 could be summarized more clearly for the reader.
- Remove line 77-78 beginning with “Several metal analyses…”
- What do the authors mean by “pooled prevalence of smoking history”?
- Define CI and OR
- Lines 99-100. The authors include studies that do not report significant associations between smoking and COIVD-19 severity; however, there could be more detail and discussion about these studies, in contrast to the studies reporting an association between smoking and COVID-19 severity.
- Section 6. Electronic cigarette, vaping, hooka and COVID-19
- Line 245. Define IQOS device
- Line 246. Remove “and bit of indirect evidence.”
- Line 252. The reference to the mouse study is missing
- Line 253. Ref 94 seems to be incorrect. Nobile and Deshusses. Biochimie 1988. This study reports on gamma-butyrobetaine uptake by the soil bacterium, Agrobacterium sp.
- Lines 258-260. The phrasing of this sentence could be revised to read as “…delivery systems could contribute to SARS-CoV-2 transmission.”
- Lines 267-270. The phrasing of this sentence could be revised to read as “…predispose smokers to respiratory infection, such as MERS-CoV.”
- Section 7. Smoking in sex predisposition and racial ethnicity in COVID-19
- Line 278. The reference for citation [105] is incorrect. It should be Haynes et al., Unmasking and addressing the toll of COVID-19 on diverse populations. Circulation. 2020;142:105–107
- Line 281-285. It would be helpful if the authors provided more discussion about ACE-2 modulation and COVID-19 resistance from the studies referenced [108 and 110].
- Line 290. The reference for citation [113] is incorrect. It should be Cai et al., Am J Respir Crit Care Med 2020; 201(12):1557-1559. This is a relevant study that the authors could spend more time discussing the analysis and findings in this review.
- Line 304. Citation [114] is in reference to a clinical trial investigating the effects of nicotine on memory improvement. This ref has no relation to the sentence on lines 303-304.
- Line 313. I was unable to find the article for citation [116] based on the information in the reference.
- Section 8. Does cessation of tobacco consumption lead to lower COVID-19 risk?
- This section deviates from the research topic and instead offers readers medical advice (lines 332-334; lines 339-341) that seems out of place.
- Section 9. Could nicotine be a therapeutic option to lower COVID-19 risk?
- Lines 368-369. The authors’ state “…complex equilibrium between ACE and ACE-2…” However, the review article only discusses ACE-2. It would be helpful if the authors provided more background on ACE and how it differs from ACE-2. This could be included in the diagram in Figure 2.
- Section 10. Bolstering defenses against COVID-19
- Lines 378-380. It would be helpful if details from citation [142] about the repurposed drugs were included. For example, the authors could specifically state that remdesivir and chloroquine were tested to evaluate inhibition of SARS-CoV-2 replication in vitro.
- Lines 381-392 and Table. This paragraph and the supporting table of natural compounds are irrelevant to the topic of smoking and COVID-19. This is out of place and would be better off removed from the article.
- Section 11. Public health strategies to reduce high-risk smoking during COVID-19
- Section 11 reads as a public health announcement.
- Figure 3 seems unrelated to the research topic discussed
- Lines 404-406. Quoting the Mayor of New York City about the increased vulnerability of smokers and vapers is out of place for this scientific review article.
- Lines 413-414. This sentence is unclear as it currently reads.
Author Response
Reviewer #2:
Kashyap and colleagues summarize current literature related to COVID-19 and the impact of smoking on disease severity. The authors highlight studies that associate smoking with changes in ACE-2 expression and pro-inflammatory cytokine responses. The authors further outline a case against smoking, as a countermeasure to the pandemic. This is an important topic relevant to the current SARS-CoV-2 pandemic. Understanding behavioral influences in the risk of COVID-19 will be important for combating disease.
Section 1. Introduction
Comment: Lines 35-36. Include specifics on the date or timeframe and countries that saw a dramatic rise in cases?
Response: Thank you very much for your advice. We included an overview of the considerable rise in COVID-19 cases from 1st February to 1st August 2020 and the most affected countries in Figure 1 in our revised manuscript in lines no of 36-38.
Comment: Line 41. Include the specific province (Guandong Province in Southeast China) in place of “Asia.
Response: We appreciate the reviewer’s suggestion. We replaced “Asia” by specifying “Guandong Province in Southeast China” in line no of 43 of revised manuscript.
Comment: Line 44. Remove the phrase “and more than 5.23 million people have recovered.” Include this phrase in a new sentence that could state that not all COVID-19 cases are fatal and include the percentage of cases that are asymptomatic.
Response: Thank you very much for your suggestion. We have removed the phrase. The paragraph was rewritten to include information on the percentage of COVID-19 cases that are asymptomatic in line no of 46-49 of revised manuscript.
Comment: Line 47. Replace “nose, mouth” with nasal and oral. End the sentence with less frequently conjunctival mucosa and begin a new sentence starting with “Common symptoms include” The authors could also include additional symptoms specific to SARS-CoV that includes loss of taste and smell
Response: We appreciate the reviewer’s suggestion. We have replaced “nose, mouth” with "nasal and oral and modified the sentences per reviewer suggestions and included additional symptoms specific to SARS-CoV in lines of 52- 55 of revised manuscript.
Comment: Line 50, revise “compared to previously health patients” to read as “compared to healthy patients without underlying medical conditions.”
Response: We appreciate the reviewer’s suggestion. We have modified the sentences per reviewer suggestions in line no 57-58 of revised manuscript.
Comment: Line 51-52. Revise influenza, H7N9 to “avian influenza H7N9 virus”
Response: We have modified the sentences as per reviewer’s suggestions in line no of 58 of revised manuscript.
Comment: Line 57-58. Revise sentence to read as, “Patients with COPD are more susceptible to viral infections, exacerbated inflammation, and loss of lung function compared to individuals with” What is meant by “those with existing exacertabing disease conditions”?
Response: Thank you very much for your valuable comments. The paragraph was rewritten and paraphrased in line no of 64-65 of revised manuscript.
Section 2. The epidemiology of smoking and COVID-19
Comment: This paragraph lists case reports for different studies without synthesizing the findings for the reader. The data reported on lines 79-98 could be summarized more clearly for the reader.
Response: Thank you very much for your valuable comments. The paragraph was rewritten and paraphrased in revised manuscript in line no of 87-115 of revised manuscript.
Comment: Remove line 77-78 beginning with “Several metal analyses…”
Response: We have removed lines 77-78 beginning with “Several metal analyses" in revised manuscript.
Comment: What do the authors mean by “pooled prevalence of smoking history”?
Response: Thank you very much for your valuable comments. The paragraph was rewritten and paraphrased in line no of 94-96 of revised manuscript.
Comment: Define CI and OR
Response: We appreciate the reviewer’s questions. We added CI (confidence interval) in line no 95 and OR (odds Ratio) in line no of 100 of revised manuscript.
Comment: Lines 99-100. The authors include studies that do not report significant associations between smoking and COIVD-19 severity; however, there could be more detail and discussion about these studies, in contrast to the studies reporting an association between smoking and COVID-19 severity.
Response: Thank you very much for your advice. We included more details in line no of 116-140 of revised manuscript.
Section 6. Electronic cigarette, vaping, hooka and COVID-19
Comment: Line 245. Define IQOS device
Response: We appreciate the reviewer’s questions. information included in line no of 290-292 of revised manuscript.
Comment: Line 246. Remove “and bit of indirect evidence.”
Response: We appreciate the reviewer’s suggestion. We have removed in revised manuscript line no of 301.
Comment: Line 252. The reference to the mouse study is missing.
Response: We appreciate the reviewer’s suggestion. We added a reference at the end of the sentences in line no of 299 in revised manuscript as reference no 101.
Comment: Line 253. Ref 94 seems to be incorrect. Nobile and Deshusses. Biochimie 1988. This study reports on gamma-butyrobetaine uptake by the soil bacterium, Agrobacterium sp.
Response: Thanks for your careful review. Line 253. Ref 94 is replaced by a new reference no 101 in line no of 301 of revised manuscript.
Comment: Lines 258-260. The phrasing of this sentence could be revised to read as “delivery systems could contribute to SARS-CoV-2 transmission.”
Response: We appreciate the reviewer’s suggestion. We have modified the sentences according to reviewer suggestions in line no 307 of revised manuscript.
Comment: Lines 267-270. The phrasing of this sentence could be revised to read as “predispose smokers to respiratory infection, such as MERS-CoV.”
Response: We have modified the sentences according to reviewer suggestions in line no 315-316 of revised manuscript.
Section 7. Smoking in sex predisposition and racial ethnicity in COVID-19
Comment: Line 278. The reference for citation [105] is incorrect. It should be Haynes et al., Unmasking and addressing the toll of COVID-19 on diverse populations. Circulation. 2020;142:105–107
Response: We replaced reference 105 as suggested by reviewer in line no 324 in revised manuscript by reference no 112.
Comment: Line 281-285. It would be helpful if the authors provided more discussion about ACE-2 modulation and COVID-19 resistance from the studies referenced [108 and 110].
Response: We have provided more details about ACE-2 modulation and COVID-19 resistance from the studies in line no of 326-342 of revised manuscript.
Comment: Line 290. The reference for citation [113] is incorrect. It should be Cai et al., Am J Respir Crit Care Med 2020; 201(12):1557-1559. This is a relevant study that the authors could spend more time discussing the analysis and findings in this review.
Response: Thanks for your careful review. Line 290. Ref 113 is replaced by references no 55 in line no 344 of revised manuscript.
Comment: Line 304. Citation [114] is in reference to a clinical trial investigating the effects of nicotine on memory improvement. This ref has no relation to the sentence on lines 303-304.
Response: Thanks for your careful review. Line 304. Ref 114 is replaced by references no 121 in line no 359 of revised manuscript.
Comment: Line 313. I was unable to find the article for citation [116] based on the information in the reference.
Response: Thanks for your careful review. Line 313. Ref 116 is replaced by a new reference no 123 in line no 368 of revised manuscript.
Section 8. Does cessation of tobacco consumption lead to lower COVID-19 risk?
Comment: This section deviates from the research topic and instead offers readers medical advice (lines 332-334; lines 339-341) that seems out of place.
Response: Thank you very much for your valuable comments. The paragraph was rewritten and paraphrased in revised manuscript in line no of 370-408.
Section 9. Could nicotine be a therapeutic option to lower COVID-19 risk?
Comment: Lines 368-369. The authors’ state “complex equilibrium between ACE and ACE-2” However, the review article only discusses ACE-2. It would be helpful if the authors provided more background on ACE and how it differs from ACE-2. This could be included in the diagram in Figure 2
Response: Thank you very much for your valuable comments. We added more information on ACE and how it differs from ACE-2 in line no of 438- 443 of revised manuscript. We also added this information in Figure. 3 in revised manuscript. Figure 2 is replaced by Figure 3 in revised manuscript.
Section 10. Bolstering defenses against COVID-19
Comment: Lines 378-380. It would be helpful if details from citation [142] about the repurposed drugs were included. For example, the authors could specifically state that remdesivir and chloroquine were tested to evaluate inhibition of SARS-CoV-2 replication in vitro.
Response: We appreciate the reviewer’s suggestion. We reviewed some of the repurposed drugs based on structure, mechanism of action, and evidence based on their reported effects in similar
viruses, so their impact on SARS-CoV-2 infection could be inferred in line no 450-514 of revised manuscript. This updated format is shown in new Figure. 4 in revised manuscript.
Comment: Lines 381-392 and Table. This paragraph and the supporting table of natural compounds are irrelevant to the topic of smoking and COVID-19. This is out of place and would be better off removed from the article.
Response: We appreciate the reviewer’s suggestion. We have removed Lines 381-392 and Table in revised manuscript as per reviewer suggestions.
Section 11. Public health strategies to reduce high-risk smoking during COVID-19
Comment: Section 11 reads as a public health announcement.
Response: We appreciate the reviewer’s suggestion. We added in line no 518 of revised manuscript.
Comment: Figure 3 seems unrelated to the research topic discussed
Response: We have removed figure 3 revised manuscript as per reviewer suggestions.
Comment: Lines 404-406. Quoting the Mayor of New York City about the increased vulnerability of smokers and vapers is out of place for this scientific review article.
Response: Per reviewer's suggestion, we have removed this quote on Lines 404-406 in revised manuscript in line no 530.
Comment: Lines 413-414. This sentence is unclear as it currently reads.
Response: The Lines 413-414 was removed in revised manuscript.